# The Dynamic Latent Block Model for Sparse and Evolving Count Matrices

Giulia Marchello [* 1 2]  Marco Corneli [* 2 3]  Charles Bouveyron [* 2]

## Abstract

We consider here the problem of co-clustering count matrices with a high level of missing values that may evolve along the time. We introduce a generative model, named dynamic latent block model (dLBM), to handle this situation and which extends the classical binary latent block model (LBM) to the dynamic case. The modeling of the dynamic time framework in a continuous time relies on a non-homogeneous Poisson process, with a latent partition of time intervals. The continuous time is handled by a time partition over the whole considered time period, where the interactions are aggregated on the time intervals of such partition obtaining a sequence of static matrices that allows us to identify meaningful time clusters. We proposed to use the SEM-Gibbs algorithm for model inference and the ICL criterion for model selection. Finally, an application with real-world data is proposed.

## 1. Introduction

In many applications, it is now frequent to have to summarize large matrices with a large amount of missing data that may evolve along the time. For instance, such data are commonly produced by e-commerce systems which record in continuous time all purchases of products made by customers. It is of great interest for those companies to cluster both customers and products to better understand the purchasing behaviors for marketing and purchase prediction. The simultaneous clustering of rows and columns of matrices is known as a co-clustering problem. We propose in this paper to add a third dimension of analysis to co-clustering by handling the dynamic of the count data generation.

---
[*]Equal contribution  [1]Department of Economics, Università degli Studi di Perugia, Perugia, Italy [2]Université Côte d'Azur, Inria, CNRS, Laboratoire J.A.Dieudonné, Maasai team, Nice, France [3]Université Côte d'Azur, Maison de la Simulation et des Interactions (MSI), Nice, France. Correspondence to: Giulia Marchello <giulia.marchello@inria.fr>.

*Presented at the first Workshop on the Art of Learning with Missing Values (Artemiss) hosted by the $37^{th}$ International Conference on Machine Learning (ICML).* Copyright 2020 by the author(s).

The latent block model (LBM) is often used as a basis for many model-based methods for co-clustering. This model was proposed for the first time by Govaert & Nadif (2003) and it is based on the assumption that rows and columns are grouped in hidden clusters. In the last decade, the model has been extended allowing to deal with counting data (Nadif & Govaert, 2010), real data (Lomet et al., 2013), categorical data (Keribin et al., 2015), ordinal data (Jacques & Biernacki, 2018; Corneli et al., 2020) and functional data (Bouveyron et al., 2018).

In this work, we extend the latent block model to the dynamic case by relying on a non-homogeneous Poisson process, with a latent partition of time intervals, for modeling the temporal data generation process. Such an approach has been already used, in a different context and with a different inference, by Corneli et al. (2018) for clustering dynamic networks. We propose here a SEM-Gibbs algorithm for model inference and to make use of the ICL criterion for model selection. The proposed model, named dynamic latent block model (dLBM), allows therefore to discover latent partitions for rows and columns, but also for the time periods. Thus, dLBM provides a meaningful summary of a massive set of data, even with a large number of missing data. We present experiments on data set extracted from Amazon to illustrate the main features of our approach.

## 2. The dynamic latent block model

In this section, we introduce the dynamic latent block model (dLBM), the main goal of this model is the simultaneous clustering of rows and columns of high-dimensional sparse matrices in a dynamic time framework. The data we consider are organized such that we denoted as $i = 1, ..., N$ the index of individuals (rows) and as $j = 1, ..., P$ the index of objects (columns). We also consider a time period [0,T] during which the total number of rows, $N$, and columns, $P$, is fixed. Moreover, we indicate as $X_{ij}(t)$ the matrix that contains the number of interactions occurring between the individual $i$ and the item $j$ at time $t \in [0, T]$. When no interaction between individuals and objects occurs, we have therefore missing values and the number of interactions is 0 for time $t$. Following Corneli et al. (2016), we further assume that the number of interactions between individuals and objects follows a non-homogeneous Poisson process

(NHPP) where the intensity function $\lambda(t)$ only depends on the clusters they belong to. The latent structure of rows and columns of the matrix $\mathbf{X}(t)$ is identified by:

- $\mathbf{z} = (z_{ik}; i = 1, ..., N; k = 1, ..., K)$ : represents the clustering of rows into $K$ groups: $\mathcal{A}_1, ..., \mathcal{A}_K$. Row $i$ belongs to cluster $\mathcal{A}_k$ iff $z_{ik} = 1$ and $\mathbf{z_i} = (z_{ik})_k \in \{0, 1\}^K$ is the group indicator of row $i$;

- $\mathbf{w} = (w_{j\ell}; j = 1, ..., P; \ell = 1, ..., L)$: represents the clustering of columns into $L$ groups: $\mathcal{B}_1, ..., \mathcal{B}_L$. Column $j$ belongs to cluster $B_\ell$ iff $w_{j\ell} = 1$ and $\mathbf{w_j} = (w_{j\ell})_\ell \in \{0, 1\}^L$ is the group indicator of column $j$;

Moreover, $\mathbf{z}$ and $\mathbf{w}$ are assumed to be independent and that they are distributed as follows:

$$p(\mathbf{z}|\gamma) = \prod_{k=1}^{K} \gamma_k^{|\mathcal{A}_k|}, \qquad (1)$$

where $\gamma_k = \mathbb{P}\{z_i = k\}; \sum_{k=1}^{K} \gamma_k = 1$ and $|\mathcal{A}_k|$ represents the number of rows in the cluster $\mathcal{A}_k$.

$$p(\mathbf{w}|\rho) = \prod_{\ell=1}^{L} \rho_\ell^{|\mathcal{B}_\ell|}, \qquad (2)$$

where $\rho_\ell = \mathbb{P}\{w_j = \ell\}; \sum_{\ell=1}^{L} \rho_\ell = 1$ and $|\mathcal{B}_\ell|$ represent the number of columns in the cluster $\mathcal{B}_\ell$. As mentioned previously, a non-homogeneous Poisson process (NHPP) is used to count the interactions between the row $i$ and the column $j$ up to time $t \in [0, T]$, denoted by $X_{ij}(t)$:

$$X_{ij}(t)|z_{ik}w_{j\ell} = 1 \sim \mathcal{P}\left(\int_0^t \lambda_{k\ell}(u)du\right), \qquad (3)$$

where $\lambda_{k\ell}(t)$ represents the intensity function that only depends on the considered row cluster $k$ and on the column cluster $\ell$. Moreover, $\lambda_{k\ell}(t)$ has to be positive and integrable on the time interval $[0, T]$.

In order to ease the understanding of the time dimension, we also assume that the whole continuous time period can be split in time clusters on which the data generation process is stable. It is worth noticing that a specific time cluster can occur more than once in the temporal line when its peculiar features are repeated after some time. Thanks to this assumption and without loss of generality the continuous time interval [0,T] can be discretized in sufficient number $U$ of subintervals $I_u = [t_{u-1} - t_u[, 0 = t_0 < t_1 < \cdots < t_U = T$, that will be then clustered. The number of interactions between

$i$ and $j$ on the considered time partition $I_u$ is summarized by $X_{iju}$ and is defined as:

$$X_{iju} := X_{ij}(t_u) - X_{ij}(t_{u-1}), \forall(i, j, u).$$

We introduce a tensor $X = \{X_{iju}\}_{iju}$ with dimensionality $N \times P \times U$. To model the membership to time clusters, a new latent variable $\mathbf{s}$ is introduced, in particular $\mathbf{s}_u = c$ if the time interval $I_u$ belongs to the time cluster $\mathcal{D}_c$. We assume that $\mathbf{s}$ follows a multinomial distribution:

$$p(\mathbf{s}|\delta) = \prod_{c=1}^{C} \delta_c^{|\mathcal{D}_c|}, \qquad (4)$$

where $\delta_c = \mathbb{P}\{s_u = c\}; \sum_{c=1}^{C} \delta_c = 1$ and $|\mathcal{D}_c|$ represents the number of time intervals in the cluster $\mathcal{D}_c$.

Once these additional assumpions have been made, we can rewrite Eq. (3) considering that the intensity functions are stepwise constant on each time cluster $\mathcal{D}_c$. Thus:

$$X_{iju}|z_{ik}w_{j\ell}s_{uc} = 1 \sim \mathcal{P}(\lambda_{k\ell c}\Delta_u) \qquad (5)$$

where $\Delta_u$ indicates the length of the interval $I_u$ that is usually constant, $\Delta_u = \Delta$. Moreover, we can set $\Delta_u = 1$ without loss of generality.

## 3. Likelihood and model inference

From Eq. (5), it holds that:

$$p(X_{iju}|z_{ik}w_{jl}s_{uc} = 1, \lambda_{k\ell c}) = \left(\frac{(\lambda_{klc})^{X_{iju}}}{X_{iju}!} \exp(-\lambda_{klc})\right) \qquad (6)$$

Therefore, we can introduce the $K \times L \times C$ tensor $\boldsymbol{\lambda}$, identified by the triplet $(i, j, u)$ and whose elements are denoted as $\lambda_{k\ell c}$. At this point it is possible to write the complete data likelihood of the model, identified by the following equation:

$$p(X, \mathbf{z}, \mathbf{w}, \mathbf{s}|\gamma, \rho, \delta, \lambda) = p(z|\gamma) \cdot p(w|\rho) \cdot p(s|\delta) \cdot p(X|z, w, s, \lambda) \qquad (7)$$

Looking at the right hand side of the Eq. (7), we can notice that $p(z|\gamma)$, $p(w|\rho)$ and $p(s|\delta)$ have been defined in the previous section respectively by the Eqs. (1), (2) and (4). The joint distribution of $\mathbf{X}$, given $z$, $w$, and $s$, can be easily obtained from Eq. (6) by independence:

$$p(X|z, w, s, \lambda) = \prod_{k,\ell,c} \left(\frac{(\lambda_{k\ell c})^{R_{k\ell c}}}{P_{k\ell c}} \exp(-|\mathcal{A}_k||\mathcal{B}_\ell||\mathcal{D}_c|\lambda_{k\ell c})\right) \qquad (8)$$

where

$$R_{k\ell c} = \sum_{i=1}^{N} \sum_{j=1}^{P} \sum_{u=1}^{U} z_{ik}w_{j\ell}s_{uc}X_{iju}$$

$$P_{k\ell c} = \prod_{i=1}^{N} \prod_{j=1}^{P} \prod_{u=1}^{U} (z_{ik} w_{j\ell} s_{uc} X_{iju})!$$

Denote by $\theta$ the set of all model parameters, i.e. $\theta = (\gamma, \rho, \delta, \lambda)$, the log-likelihood can be written as:

$$\ell(\theta; X) = \sum_{z} \sum_{w} \sum_{c} \log p(X, z, w, s | \theta) \qquad (9)$$

### 3.1. Model inference

As usual, we look for a way to maximize the log-likelihood in order to obtain the estimation of $\boldsymbol{\theta}$. In the co-clustering case, the EM algorithm is computationally infeasible. In this work, we have chosen to use a different inference strategy than the one used in Corneli et al. (2016) that relies on a maximization, through a greedy search algorithm, of the derived exact integrated classification likelihood criterion. The inference strategy we use here for dLBM is known as SEM-Gibbs, proposed by Keribin et al. (2010) and exploited, for instance, by Bouveyron et al. (2018) in the functional latent block model (funLBM). During the SE step the algorithm evaluates the posterior probabilities using the current values for the parameters, while during the M step a new estimation of the model parameters is made. Thanks to the Gibbs Sampling, in the SE step a partition for $\boldsymbol{z}$, $\boldsymbol{w}$ and $\boldsymbol{s}$ is generated without computing the joint distribution. The algorithm starts with initial values for the parameter set $\boldsymbol{\theta}^{(0)}$, the column clusters $\boldsymbol{w}^{(0)}$ and the time clusters $\mathbf{s}^{(0)}$. Regarding the burn-in period, after a certain number of iterations of the algorithm, we can obtain the final parameters estimation by computing the mean of the sampled distribution. The optimal values for $\boldsymbol{z}$, $\boldsymbol{w}$ and $\boldsymbol{s}$ are estimated by the mode of their sample distributions.

### 3.2. Model selection

Up to now, we have assumed that the exact number of row clusters ($K$), column clusters ($L$) and time clusters ($C$) to be included in the algorithm were known. However, for real data sets this assumption is unrealistic. For this reason, our purpose in this section is to define a model selection criterion that can automatically identify the optimal number of clusters that compose a data set. The model selection approach is considered. We propose to rely on the ICL (Integrated Completed Likelihood, Biernacki et al. (2000)) criterion to approximate the complete-data integrated log-likelihood. We derived the formulation of the ICL criterion for the model proposed above:

$$ICL(K, L, C) = \log p(X, \hat{z}, \hat{w}, \hat{s}; \hat{\theta}) - \frac{K-1}{2} \log n +$$
$$- \frac{L-1}{2} \log p - \frac{C-1}{2} \log u - \frac{KLC}{2} \log(npu) \qquad (10)$$

The triplet $(\hat{K}, \hat{L}, \hat{C})$ that leads to the highest value for the ICL is considered as the most correct for those data.

## 4. Analysis of the Amazon fine foods dataset

This section focuses on a real dataset consisting of reviews of fine foods from Amazon. The dataset can be freely downloaded at https://snap.stanford.edu/data/web-FineFoods.html. A time horizon of 10 years is considered, up to October 2012. In the original dataset, the number of reviews reported is 568,464 and each row corresponds to one review. Some additional information is reported for each review: the user/product numerical identifiers, a summary of the review and a rating attributed to the product by the user. The rating is expressed via an integer number spanning from 1 (very bad) to 5 (very good). To focus on the most meaningful part of the data, we only considered the users reviewing more than 30 times and the products being reviewed more than 55 times.

In the Amazon Fine Foods dataset, we consider one month as the dimension of a single time interval as $\Delta_u$. The distribution of the observations during the considered time period is shown in Figure 1. To be able to run our algorithm, we need to group the data building a sparse array with $N$ rows, $P$ columns and $U$ slices. Thus, we end up with a $235 \times 151 \times 52$ cube $X$, where each user is represented by a row, each product is represented by a column and each time interval is represented by a slice of the cube. Each entry of the cube represents an interaction between a reviewer and a product on a specific time interval. In the final data set, the rate of missing values is 0.98%.

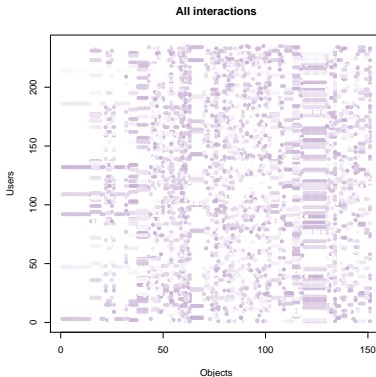

*Figure 1.* Distribution of users and products before running the algorithm

Once the array has been built, we can run the model using the SEM-Gibbs algorithm to identify the optimal number of $K$ row clusters (users), $L$ column clusters (products) and $C$ time clusters. We run the algorithm over 48 different models, with rows and columns groups ranging from 2 to 5 and time clusters groups ranging from 2 to 4. To run this

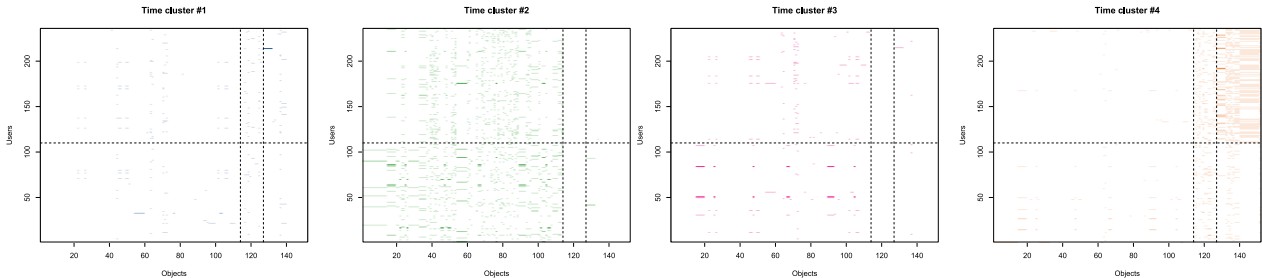

*Figure 2.* Reorganized incidence matrix for each time cluster. Rows and columns clusters are delimited by the dashed lines while the colored dots marks an interaction (i.e. review) between a user and a product.

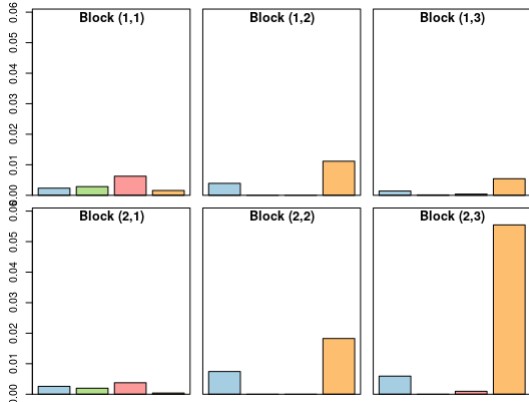

*Figure 3.* Estimated parameters of the probability distribution of interactions between users and products, according to time clusters

experiment the algorithm takes approximately 6 hours. As shown in Table 1, the ICL selects as the best dLBM model the one with 2 groups of users, 3 groups of products and 4 time clusters.

| Rank | K | L | C | ICL |
|------|---|---|---|-----------|
| 1 | 2 | 3 | 4 | -32509.56 |
| 2 | 2 | 5 | 2 | -32916.66 |
| 3 | 2 | 4 | 2 | -32919.41 |
| 4 | 4 | 5 | 2 | -32957.24 |
| 5 | 4 | 2 | 2 | -32978.14 |

*Table 1.* Selection of the most appropriate model for the Amazon Fine Foods data.

In Figure 2, one can observe the reorganized incidence matrix according to row and column clusters, and this for each time cluster (each panel represents a different time cluster). To this end, we permuted the rows and the columns of the incidence matrix thanks to the estimates $\hat{z}$ and $\hat{w}$ in such a way that nearby rows (columns) belong to the same cluster of rows (columns). The blocks are also delimited by the black dashed lines.

One can notice that time cluster 1 contains few and sparse

interactions, without a clear structure regarding the blocks. This means that, for time periods belonging to this time cluster, the users comment the product on an irregular basis and almost randomly. Conversely, in time cluster 2, there is a clear structure regarding the blocks: all users have a clear tendency to comment and buy products of the first column clusters whereas they do not consider at all products of the two other column clusters (2 and 3). It can be also noticed that users of the firs row cluster (bottom row blocks) have a higher probability to comment products of the first column cluster than users from the second row cluster. The third time cluster is quite similar to the 2nd time cluster in term of block structure, but the interaction intensity is clearly lower. Finally, time cluster 4 shows a very specific structure since, this time, the first group of product does not receive much attention whereas the products of the second column cluster are frequently commented by all users and the ones of the third column cluster are very intensively reviewed by the users of the 2nd row cluster (top row block). These analyses are also summarised in Figure 2 which shows the estimated valued for the tensor $\boldsymbol{\lambda}$.

## 5. Conclusion

We considered the problem of co-clustering count matrices with a high level of missing values that may evolve along the time. To this end, we introduce a generative model, named dynamic latent block model (dLBM), which extends the classical binary latent block model to the dynamic case. The modeling of the dynamic time framework in a continuous time relies on a non-homogeneous Poisson process, with a latent partition of time intervals. Inference is done using a SEM-Gibbs algorithm and the ICL criterion is used for model selection. The dLBM algorithm was applied to a real-world data set from Amazon with a extremely high level of missing values. In this context, dLBM provided meaningful segmentation of rows, columns and time.

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
