# OpenReview forum: "The Dynamic Latent Block Model for Sparse and Evolving Count Matrices"
_ICML.cc/2020/Workshop/Artemiss — ICML Artemiss 2020_

### Official Review · AnonReviewer2 · 2020-06-30
**The paper has some shortcomings, but it is a good workshop paper.**

**Confidence:** 4
**Rating:** 7

**Review:**

The authors propose a latent block model for sparse count matrices that evolve temporally. In this context, a 0-entry in the count matrix is considered a missing value. Similar to Corneli et al. (2016), the entries in the matrix is assumed to follow a non-homogeneous Poisson process, where the rate parameter is a tensor with one value for each combination of column cluster, row cluster and temporal cluster. The main novelty is the introduction of the temporal cluster. Model inference is done using Stochastic EM with Gibbs sampling, as in previous work, and model selection is made using ICL. The proposed method is applied to the Amazon Fine Foods dataset, and it is qualitatively assessed that the segmentation is meaningful.

The paper proposes an interesting temporal extension to the collection of latent block models. The main drawback of the paper is that it does not discuss existing literature on temporal co-clustering of (count) matrices (if such literature exists?). Furthermore, a more quantitative evaluation of the method would also be desirable (e.g. just a simple experiment that tests how robust are the number of clusters are estimated using data generated from the model with known numbers of clusters).

Despite these shortcomings, I think the paper is very well suited as a workshop paper.

Minor comments:
* The definition of $\gamma_k = \mathbb{P}(z_i=k)$ after equation (1) is inadequate. Partly because the value of $i$ is not specified and partly because $z_i$ is not defined (only $\mathbf{z_i}$, which is a vector, and $z_{ik}$, which is a binary value, is define). Therefore, it is also not clear that $p(\mathbf{z}|\gamma)$ is well-defined (e.g. it is not clear to me that $\sum_\mathbf{z} p(\mathbf{z}|\gamma) =1 $). The same is the case for $\rho_\ell$ in equation (2) and $\delta_c$ in equation (3).
* There seems to be missing some text before equation (2).
* In equation (3), I would put parentheses around $z_{ik}w_{j\ell} = 1$, as this would make the equation easier to read (otherwise the reader has to be aware that $\sim$ take precedence over $=$). The same goes for equation (5).
* I believe that $z$, $w$ and $s$ should be bf on the RHS of equation (7) and LHS of equation (8).
* In section 3.1, I would recommend writing out the abbreviation SEM and also referencing the original paper (Celeux and Diebolt, 1985).

---

### Decision · Program_Chairs · 2020-07-02

**Decision:**

Accept

**Comment:**

We are very happy to inform you that your paper has been accepted for the Artemiss workshop. We will contact you soon to inform you about the details concerning the format of your presentation at the workshop, and the camera-ready version deadline. Please take into account the referee's comments to write the camera-ready version.